# Characterisation of the Gastrointestinal Microbiome of Green Sea Turtles (*Chelonia mydas*): A Systematic Review

**DOI:** 10.3390/ani15111594

**Published:** 2025-05-29

**Authors:** Dawood Ghafoor, Orachun Hayakijkosol, Carla Ewels, Robert Kinobe

**Affiliations:** 1Veterinary Preclinical Sciences, College of Science and Engineering, James Cook University, Townsville, QLD 4811, Australia; dawood.ghafoor@my.jcu.edu.au (D.G.); orachun.hayakijkosol1@jcu.edu.au (O.H.); 2Centre for Tropical Biosecurity, James Cook University, Townsville, QLD 4811, Australia; 3Statistics and Data Sciences, College of Science and Engineering, James Cook University, Townsville, QLD 4811, Australia; carla.ewels@jcu.edu.au; 4Centre for Molecular Therapeutics, Australian Institute of Tropical Health and Medicine, James Cook University, Townsville, QLD 4811, Australia

**Keywords:** green sea turtle, *Chelonia mydas*, gut microbiome, conservation, microbial diversity, captivity, turtle rehabilitation, wild turtles, marine ecosystem

## Abstract

Gut bacteria are a critical determinant of health, but they are not fully defined in green sea turtles (*Chelonia mydas*). This review determined what constituted dominant gut bacterial phyla in green sea turtles; probable microbial shifts between wild and captive populations were identified. These potential microbial shifts that are likely to be shaped by environmental factors are important in guiding management, rehabilitation, and conservation of green sea turtles.

## 1. Introduction

Green sea turtles (*Chelonia mydas*) are important marine reptiles found in tropical and subtropical oceans worldwide [1]. They are considered indicators of marine environmental health due to their long lifespans and extensive migratory patterns [2,3,4]. Green sea turtles play a vital role in ocean ecosystems by maintaining the health of seagrass beds and coral reefs [5]. Despite their ecological significance, the International Union for Conservation of Nature (IUCN) classified green sea turtles as an endangered species due to several anthropogenic and environmental threats [3,5]. As a result, a decline in the green sea turtle population would destabilise marine ecosystem health.

Green sea turtles undergo extensive migrations between natal beaches, pelagic habitats, and coastal foraging grounds and exhibit a complex life cycle with several distinct developmental stages starting with hatchlings and progressing through juvenile, sub-adult, and adult stages [6]. Their diet transitions from primarily carnivorous as hatchlings, consuming small invertebrates, to predominantly herbivorous as adults, feeding mainly on seagrasses and algae [7,8]. Green sea turtles are exposed to various factors that can significantly impact their health throughout their life cycle. These factors include impactful changes in environmental conditions such as water temperature and quality, food availability, and habitat degradation. Exposure to pollutants, pathogens, and climate change can also exacerbate health issues and impact green sea turtles’ overall fitness and life expectancy [9,10].

The composition and functionality of the gut microbiome are critically associated with the health of green sea turtles [11,12]. In animals, the gut microbiome is a diverse and dynamic community of microorganisms residing in the gastrointestinal tract, and it plays a critical role in their health by facilitating nutrient digestion, modulating immune system development and function, and regulating physiological homeostasis [13,14,15,16,17]. Gut microorganisms can increase the metabolic potential of the host by aiding the utilisation of complex, indigestible food particles like polysaccharides [13,16]. Commensal bacteria enhance host immunity through pathogen invasion, stimulation of host antimicrobial responses, and regulation of immune cell development and differentiation [18]. Multiple factors can affect the gut microbial composition and diversity in green sea turtles, and these may include diet, age, physiological conditions, habitat characteristics, and exposure to antibiotics and other xenobiotics [11,19,20,21,22]. Environmental stressors such as pollutants, contaminants of emerging concern, and climate change can further disrupt the stability of gut microbiomes [23]. Disruptions in the gut microbial community have been associated with the dysregulation of immune responses, metabolic pathways, and overall physiological health, which can result in increased susceptibility to diseases, inadequate nutrient absorption, and alterations in physiological functions [24,25].

Given the importance of the microbiome in influencing the health of green sea turtles, a comprehensive characterisation of the gut microbiome of green sea turtles is crucial. An earlier review provided an overview of the gut microbiome across all sea turtle species [26]. However, this previous review incorporated multiple turtle species with limited focus on any one specific marine turtle species, and it lacked in-depth analysis of factors that affect gut microbiomes in green sea turtles. In the current study, we conducted an updated systematic review to deepen insights into the gut microbiome of green sea turtles from both wild and captive environments. The specific aim was to determine the dominant bacterial phyla in the gut microbiomes of wild and captive green sea turtles and to elucidate factors that are associated with gut microbial changes. It is predicted that the results from this review will provide preliminary insights into the gut microbial community composition of green sea turtles, and this may guide conservation and health management by veterinarians and conservationists in the field.

## 2. Methodology

### 2.1. Searching Strategy

A systematic search of the literature was performed according to the Preferred Reporting Items for Systematic Reviews and Meta-Analyses (PRISMA) guidelines [27]. Three databases, including Medline (Ovid), Scopus, and Web of Science, were considered because of their likelihood of containing information on the proposed research question. To enhance the likelihood of capturing all relevant literature in these three databases, the search terms used were (“sea turtle” OR “sea turtles” OR “marine turtles”) AND (“microbial communities” OR “microbial community composition” OR “microbiome” OR “microbiota”). This allowed for the identification of the documents available in databases on the gut microbiome of sea turtles. The term “sea turtle” was used in the search strategy to ensure broader coverage and include studies where green turtles were studied alongside other species. All used databases were searched from the time of inception to 25 April 2024. All relevant articles were accessed via institutional subscriptions and open-access repositories.

### 2.2. Inclusion and Exclusion Criteria

To align with the primary search objective, eligibility was limited to primary data sources in journal articles published in English. The exclusion criteria included papers that did not focus on the gut microbiome of marine turtles, did not specifically investigate the gut microbiome of green sea turtles, lacked phylum-level data on microbial dominance, or were review articles. Research articles deemed to be relevant to the topic were further assessed and individually qualified for data extraction. Each article was first screened for eligibility based on its title, abstract, full text, and references to exclude irrelevant studies and remove duplicate records. Only articles meeting the inclusion criteria were subsequently qualified for data extraction, where key information was systematically collected for further analysis. The inclusion criteria and the extracted information were used to design a ten-point scoring system for the qualitative evaluation of individual publications, as previously outlined [28]. The article scoring scheme used here is shown in the Appendix A (Appendix A).

### 2.3. Categorisation and Data Extraction

#### 2.3.1. Categorisation of Animals

Animals were categorised based on their geographical locations to assess where the gut microbiome of green sea turtles was studied. Animals and obtained samples were designated as wild if samples were obtained from turtles in their natural marine habitats or upon their arrival at rehabilitation centres, while captive was defined as animals and samples collected from sea turtles housed in captivity for more than two months. The health status was evaluated and categorised according to the focus of each study, with turtles classified as either healthy or non-healthy by the respective study authors.

#### 2.3.2. Data Extraction

Data were extracted from studies focusing on the gut microbiome of green sea turtles. Additionally, published studies for other purposes but containing relevant gut microbiome data on green sea turtles were included to provide a comprehensive overview. The gut microbiome data were meticulously extracted and categorised from text, figures, and tables, emphasising crucial variables such as species, sampling timelines, location, interventions, rehabilitation status, and sample sizes as presented in the Appendix A.

#### 2.3.3. Data Analysis

Extracted data were collated and presented as qualitative descriptions. The relative abundance of phyla across studies was ranked, and the mode of these ranked values was calculated to identify the predominant phyla. For direct comparisons, the ranks and modes for different phyla were stratified into rational groups, including wild versus captive and geographical location. The marked heterogeneity in experimental conditions and data generated across all eligible studies prevented any formal meta-analyses or detailed statistical evaluations.

## 3. Results and Discussion

### 3.1. An Overview of Qualitative and Quantitative Features of Evaluated Studies on the Gut Microbiome of Green Sea Turtles

A systematic search across Medline (Ovid), Scopus, and Web of Science identified 129 published articles (29 from Medline (Ovid), 49 from Scopus and 51 from Web of Science). Notably, all the identified articles were published in the English language. A total of 64 duplicate articles were eliminated, and 65 individual articles were reviewed. Of these, 52 additional articles were excluded for not meeting all the inclusion criteria with reasons, and detailed analyses were based on 13 peer-reviewed primary articles. A detailed PRISMA schematic flow chart and evaluation of all retrieved articles are provided in Figure 1.

### 3.2. Represented Geographical Locations in Studies on Gut Microbiome of Green Sea Turtles

The global distribution of marine turtles is extensive, except for Arctic and Antarctic waters [29]. Their highly migratory nature allows them to traverse foraging areas that often span thousands of kilometres during periodic migrations every few years [30]. Herein, however, it is shown that peer-reviewed, published research on the gut microbiome of green turtles (including both captive and wild) is currently limited to five countries. These include Australia, the United States of America (USA), China, Brazil, and Guinea-Bissau. Five studies have been conducted in Australia, specifically focusing on the Coral Sea regions situated off the northeast coast of the country [11,17,21,31,32]. These locations are known for their abundant marine life and are particularly famous for their vast coral reefs, including the renowned Great Barrier Reef. In the USA, three studies have been carried out, covering specific locations, including the Georgia Sea Turtle Center, the Northern Gulf of Mexico, St. Joseph Bay, Crooked Island Sound, St. Andrews Bay, and Santa Rosa Island [19,33,34]. Research in China has been represented by three studies centring on the Sea Turtles National Nature Reserve in Huidong within Guangdong Province [35,36,37]. In Brazil, one study focused on the coastal waters around Praia do Forte and Ubatuba [20], while Guinea-Bissau contributed one study covering the islands of Unhocomo, Unhocomozinho, Poilão Island, and the Bijagós Archipelago [12]. The distribution of studies across geographic locations, along with the dominant phyla identified in captive and wild green sea turtles at these sites, is summarised in Table 1.

### 3.3. Comparative Analysis of Gut Microbiome in Wild and Captive Green Sea Turtles

Of the thirteen studies included in detailed analysis for this review, four focused on captive green sea turtles, one on both wild and captive individuals, and the remaining eight collected samples from wild animals. Notably, all captive green sea turtles were juveniles in good health, whereas the wild population included healthy, stranded, and deceased turtles. It was apparent that reporting of research data on gut microbial communities in turtles is not standardised. While some reports included bacterial class and genera [11,17,19,21,31,33,34,35,36,37], only bacterial phyla were captured consistently. In this review, therefore, the authors focused on reported gut bacterial phyla, and commentary on class, genera, or species was provided as identified in individual studies. Bacillota (formerly Firmicutes), Bacteroidota (formerly Bacteriodetes), Verrucomicrobiota (formerly Verrucomicrobia), and Pseudomonadota (formerly Proteobacteria) were consistently identified, at relatively higher abundances in the microbiome of green sea turtles in all included studies; these made up the important focal phyla for detailed analysis (Figure 2). Other phyla were either absent, present at very low levels, or rarely reported.

This demonstrates a likely dominant role of these top four phyla in structuring the microbial composition, while other phyla, including Actinomycetota (formerly Actinobacteria), Fusobacteriota (formerly Fusobacteria), Spirochaetota (formerly Spirochaetes), Absconditabacteria (formerly SR1), Cyanobacteriota (formerly Cyanobacteria), Lentisphaerota (formerly Lentisphaerae), and Candidatus Saccharibacteria (formerly TM7), were less abundant. Appendix A provides a visual distribution of the relative abundances for less abundant phyla across studies that were evaluated for this review.

#### 3.3.1. Bacillota

Analysis in this study showed that Bacillota was the predominant phylum in captive green sea turtles, with relative abundances ranging from 40.9 to 87.5% [31,35] (Figure 2). In contrast, wild populations exhibited greater variability in the Bacillota phylum, with relative abundance ranging from 3.5 to 57.8% [17,33] (Figure 2). This difference suggests that the more stable and controlled environment of captivity likely supports a consistent microbiome, with dietary management and standardised husbandry practices contributing to the high relative abundance of Bacillota [38]. The relatively high and stable abundance of Bacillota in healthy captive turtles is consistent with the general suggestion that captivity may provide an optimal healthy environment for green sea turtles [26]. In contrast, the variability observed in wild populations may be influenced by environmental factors in the wild, such as variability in food availability and exposure to pathogens and pollutants. Notably, within wild populations, higher abundances of Bacillota were predominantly associated with healthy turtles, while stranded and deceased turtles exhibited greater variability in this bacterial phylum [12,19,20,33,34,35,36,37]. These fluctuations may reflect the significant role of ecological, dietary, and health-related factors in shaping the gut microbiome, and this has been suggested by other scholars [17,31,39,40]. Bacillota has also been observed in other marine species in captivity, which indicates its ecological significance in marine ecosystems. For instance, Bacillota was the predominant phylum in the gut microbiomes of loggerhead and hawksbill sea turtles, as reported previously [36,41]. Similarly, marine mammals such as beluga whales, Pacific white-sided dolphins, and Cape fur seals revealed a high abundance of Bacillota in their gastrointestinal microbiota [42]. This consistent pattern across diverse marine species reflects the fact that Bacillota may serve a critical role in maintaining normal health and microbial stability in marine animals.

Within the phylum Bacillota, Bloodgood et al. [19] reported that the class *Clostridia* dominated the gut microbiome of wild green sea turtles, comprising 51% of the microbial community. This class was primarily represented by the families Lachnospiraceae (12%), Bacteroidaceae (Bacteroides, 10.7%), and Ruminococcaceae (6.1%). Ahsan et al. [17] also identified *Clostridia* as a predominant class in stranded wild green sea turtles, but its relative abundance was lower (34.1%). Among the genera within *Clostridia, Clostridium sensu stricto 1* was particularly prominent, accounting for 41.5% of the microbiome in captive turtles [31]. Similarly, *Clostridium* was prevalent in wild-captured sea turtles, where it accounted for 18.3% of the microbiome [21]. Other detected genera within Clostridia included *Peptoclostridium* (7.2%) and *Lachnospiraceae* UCG-004 (1.4%) in captive green sea turtles [31]. In pre-hospitalisation samples, *Clostridium sensu stricto 1* (4.4%) and *Peptostreptococcus* (5.6%) were dominant genera within *Clostridia* [11], and other represented genera within Bacillota included *Blautia* (3.9%) and *Paludibacter* (10.3%) [35,36]. The class *Erysipelotrichia* was also represented within Bacillota, though at a much lower relative abundance, as shown by [19], where it accounted for 0.2%.

Members of Bacillota are adept at breaking down complex carbohydrates such as cellulose, hemicellulose, and xylan found in seagrasses and other plant fibres [43,44]. Nonetheless, several genera within the phylum Bacillota were also reported in clinical infections. For example, *Peptostreptococcus* and *Eubacterium* identified in the pre-hospitalisation samples were reported in clinical infections in immune-compromised hosts [45,46]. It is noteworthy that a significant proportion of *Peptostreptococcus* was found in turtles, as well as *Clostridium botulinum*, which produces botulinum, the neurotoxin that causes botulism in humans and animals [47]. These observations suggest that some bacterial classes and genera within the Bacillota phylum may proliferate under disease conditions. Therefore, it can be suggested that future research should focus on species-level identification of Bacillota and determine their exact role within the gastrointestinal tract (GI tract) of green sea turtles.

#### 3.3.2. Bacteroidota

Bacteroidota was a consistently prevalent phylum in the gut microbiome in green sea turtles across captive and wild groups. However, its relative abundance showed marked variability between these two populations. In captive green sea turtles, the relative abundance of Bacteroidota ranged from 8.7 to 45.6%, ranking as the second most dominant phylum within the microbiome [20,31] (Figure 2). Other studies recorded relative abundances of 39.7% [37], 11.8% [36], 39.5% [35], and 29.6% [21]. In wild green sea turtles, the abundance of Bacteroidota varied widely, ranging from 3.6 to 43.1% [17,20] (Figure 2), and the highest abundance (43.1%) was reported by Campos et al. [20] in both healthy and deceased turtles. Similarly, this marked variability in the relative abundance of Bacteroidota in wild and captive green sea turtles is likely influenced by differences in the immediate environments inhabited by these animals. To support this notion, it has been suggested that the provision of stable diets and reduced exposure to ecological stressors may promote a richer and uniform microbial community [36,37,38]. To provide an example, Ahsan et al. [21] reported a lower abundance of Clostridia (19.0%) in stranded green sea turtles and suggested this microbial change may have been linked to disrupted feeding patterns or compromised health [21]. Similarly, Forbes et al. [34] found that Bacteroidota had a relative abundance of 19.3% in deceased turtles; these authors also suggested that health or environmental factors had affected the microbial composition. Nonetheless, empirical evidence to support all these associations is lacking; this should be the focus of future studies in this field.

Within Bacteroidota as a phylum, *Bacteroidia* was a prominent class, and it has been suggested that this class is vital in shaping the microbial diversity and functional dynamics of the gut microbiome of green sea turtles [26,48]. This group of bacteria degrades complex polysaccharides, facilitates nutrient absorption, and contributes to the host’s overall health [49]. The summary provided in this review indicates that there are significant variations in the relative abundance of *Bacteroidia* across different environmental contexts. For instance, Ahsan et al. [17] found that Bacteroidia represented approximately 1.4% of the total microbial community in green sea turtles. Other studies reported much higher abundances, with Bacteroidia comprising 25.1% of the microbiome in wild-captured turtles and 12.6% in stranded individuals [21]. In contrast, the relative abundance of *Bacteroides* was noticeably lower in captive turtles, where it accounted for only 7.8% of the microbiome [31]. *Parabacteroides* was among the notable genera within Bacteroides with relative abundances ranging from 0.1 to 5.7% across different studies [19,31]. While the functional role of this genus in sea turtles remains unknown, its ecological relevance has been noted in other aquatic animals. In zebrafish, for instance, *Parabacteroides distasonis* was found to regulate the infectivity and pathogenicity of Spring viremia of carp virus under varying water temperatures [50]. Other notable genera in *Bacteroides* included *Macellibacteroides*, which exhibited relative abundances of 9.9% in wild-captured turtles and 9.8% in stranded individuals [11,21]. *Macellibacteroides* are common gut-associated microbes reported in several aquatic and terrestrial vertebrates, including herbivorous mammals such as the dugong [51,52,53]. *Macellibacteroides* are associated with the decomposition of cellulose- and hemicellulose-derived sugars [54,55]. By comparison, *Tenacibaculum* seemed to vary with the specific location of wild green sea turtles in marine environments with relative abundances of 2.0% in pelagic, 5.1% at beachfronts, and 0.9% in bays [33]. Similarly, this genus has not been associated with any specific pathophysiological roles within the GI tract of green sea turtles. Nonetheless, some species in this genus, for example, *Tenacibaculum maritimum*, are pathogenic and responsible for tenacibaculosis, which affects marine fish, causing significant economic losses in aquaculture [56].

#### 3.3.3. Verrucomicrobiota

Verrucomicrobiota was less prevalent in wild and captive populations, with relative abundances ranging from 0.3 to 5.4% and 2.3 to 7.2%, respectively [31,36] (Figure 2). A genomic analysis of predominant polysaccharide-degrading Verrucomicrobiota revealed a broad array of genes that encode for enzymes such as glycoside hydrolases and sulfatases, which confirms their extensive capability for polysaccharide hydrolysis [57]. Common representatives of Verrucomicrobiota are bacteria in the *Akkermansia* genus, and these are mucin-degrading bacteria, commonly found in the human gut and have also been isolated in other reptiles [58,59]. *Akkermansia* species are linked to gut health benefits, such as enhancing gut barrier function and reducing inflammation in humans and mice [60,61], but may compromise gut integrity under low-fibre diets [62]. Nothing is known about their pathogenicity in reptiles.

In the current study, the relative abundances of *Akkermansia* varied from 1.5 to 4.7% in captive turtles [31,36] and it was 10.3% at the genus level in wild turtles [19]. It was not possible to establish, however, whether *Akkermansia* and any related genera were associated with significant pathophysiological changes in the wild or captive green sea turtles that were evaluated.

#### 3.3.4. Pseudomonadota

Pseudomonadota was the fourth most dominant phylum in captive green sea turtles, with relative abundances ranging from 0.1 to 6.6% (Figure 2). In contrast, wild turtles exhibited a significantly higher relative abundance of Pseudomonadota, in which it ranked as the most predominant phylum, with relative abundances ranging from 6.2 to 68.1% [19,20,21,32,33,34] (Figure 2). Notably, the upper end of the range observed in captive turtles (6.6%) is within the range observed in wild turtles. These data suggest a relatively higher presence of Pseudomonadota in wild populations, which could be attributed to the natural diversity and ecological pressures in the wild compared to the more controlled conditions of captivity. Due to the heterogeneity in field conditions and health status of evaluated wild green turtles in the current study, the relatively higher proportion of Pseudomonadota was not directly correlated with any pathological features. Elucidating these links should be the focus of future experimental and observational studies.

Pseudomonadota was predominantly represented by *Gammaproteobacteria*, and this class accounted for 13.1% of the microbiome in a study by Bloodgood et al. [19]. In comparison, other classes within Pseudomonadota were present at relatively low proportions, with *Alphaproteobacteria* comprising 0.1% and *Deltaproteobacteria* contributing 1.7% [19]. Similarly, in the study by Ahsan et al. [17], *Gammaproteobacteria* represented 19.7% of the microbiome, while *Alphaproteobacteria* made up 3.2%, and this emphasises the dominance of *Gammaproteobacteria* in green sea turtles.

At the genus level within Pseudomonadota in captive green sea turtles, *Paracoccus* 6.5%, *Acinetobacter* 3.4%, and *Pseudomonas* 0.4% were identified among the predominant genera [31]. In wild turtles, a higher microbial diversity was observed, and *Citrobacter* 26.2%, *Psychrobacter* 1.4%, and *Salmonella* 11.3% were identified as the dominant genera [37]. In stranded turtles, *Escherichia–Shigella* 7.5%, *Psychrobacter* 6.2%, and *Providencia* 5.9% were the most prevalent [21]. *Epsilonproteobacteria* were associated with pre-hospitalisation of green sea turtles, while *Deltaproteobacteria* predominated in post-rehabilitation samples [11]. In addition, some habitat-specific variations in genera were also observed with *Moraxella* at 15.9% and *Arcobacter* at 5.5% in beachfront samples and 5.6% in bay samples [33]. This emphasises the influence of environmental factors on gut microbiomes in green sea turtles.

More generally, Pseudomonadota is often associated with environmental stress and dietary fluctuations [19,63]. A relatively higher proportion of Pseudomonadota in wild green sea turtles may indicate that this group of animals might have experienced more variable conditions that affected their gut microbiome. This notion is plausible because a high prevalence of Pseudomonadota within the gastrointestinal tract is a recognised signature of dysbiosis and an indication of disease in humans and animals [20,64,65]. However, because of the phylogenetic diversity in Pseudomonadota, this phylum could also sub-serve physiological or biochemical functions for the host animal. Pseudomonadota can play a vital role in preparing the juvenile gut for transition to adulthood through oxygen consumption, altering the gut pH, and producing carbon dioxide and nutrients for successive colonisation by strict anaerobes [66,67]. On the other hand, members within this phylum have also been reported to establish pathogenic relationships with their hosts [68,69,70]. Bacterial species such as *Campylobacter*, *Arcobacter*, *Escherichia*, *Edwardsiella*, *Citrobacter*, and *Shewanella* were reported to cause opportunistic infections in several terrestrial and aquatic vertebrates, including sea turtles [71,72,73]. These bacterial species are often associated with lesions and stress, contribute to severe disease conditions such as carapacial ulcers, pneumonia, and septic arthritis and have been linked to mortalities [70]. *Psychrobacter* has been linked to carapacial ulcers [37], while *Acinetobacter* spp. has been found in leatherback hatchlings’ blood samples and linked to lethal damage [74]. In addition, species within *Pseudomonas*, including *P. aeruginosa*, *P. fluorescens*, and *P. putrefaciens*, have been isolated from sea turtles with lesions for fibropapillomatosis and pulmonary disease [70]. This highlights the fact that *Pseudomonas* species also play a role as opportunistic pathogens in infections that may be exacerbated by environmental trauma and stress [63,75].

### 3.4. Comparative Analysis of the Gut Microbiome in Captive and Wild Green Sea Turtles Across Health Conditions

A comparative analysis of the gut microbiota in wild and captive *Chelonia mydas* revealed distinct community compositions associated with both habitat and health status. In captive healthy individuals, Bacillota predominated, ranging from 40.9% to 87.5%, followed by Bacteroidota (8.7% to 45.8%), Verrucomicrobiota (2.3% to 7.2%), and Pseudomonadota (0.1% to 5.7%) [20,31,35,36,37]. In contrast, captive unhealthy turtles exhibited a substantial shift toward Pseudomonadota (47.6% to 73.2%), accompanied by decreased relative abundances of Bacteroidota (13.2% to 19.0%), Bacillota (9.9% to 18.7%), and Fusobacteriota (0% to 13.6%) [21,37].

Wild turtles that were reported as healthy displayed greater microbial diversity with Bacillota ranging from 3.5% to 61.6%, Bacteroidota (11.4% to 44.4%), and Pseudomonadota (5% to 68.1%). Other phyla, including Verrucomicrobiota (0% to 3.1%) and Actinomycetota (0% to 17%), were present in relatively lower proportions [12,19,20,32,33,76]. In wild turtles reported as unhealthy, Bacillota were dominant (25.5% to 57.8%), followed by Pseudomonadota (21.3% to 33.7%), Bacteroidota (3.6% to 14.4%), Fusobacteriota (2.4% to 9.1%), and Actinomycetota (0.4% to 6.4%) [11,17]. Turtles that were found dead were primarily colonised by Bacillota (51%) and Bacteroidota (40.6%), but one study reported a pronounced increase in Pseudomonadota (50.8%) [20,34]. The relative abundances of dominant and other notable minor phyla across health categories are presented in Figure 3A,B, while phyla with very low relative abundance are detailed in Appendix A.

## 4. Limitations of the Study

This study was limited by the inability to stratify all generated data according to key variables such as diet, age, geographic origin, ecological background and bacterial phyla, class or genera. These factors are crucial for understanding the complex interactions between host physiology, environmental influences, and the microbial composition. This is primarily because very limited data exist in this field, and thus, a systematic search of the literature yielded a relatively small number of studies, with marked heterogeneity. As a result, meta-analyses and detailed statistical evaluations were beyond the scope of this study. Moreover, data regarding the underlying diseases responsible for stranding or illness, the severity and extent of these conditions, and any administered antibiotic treatments were not reported in the included studies. These variables are likely to influence the gut microbiome and represent critical gaps in the current study. Additionally, this is a systematic review; newly published studies after our cut-off date could offer more insights and should be considered in future research. Nonetheless, this review summarises current knowledge relating to gut microbiomes in captive and wild green sea turtles, thus providing a foundation for future studies.

## 5. Conclusions and Future Perspectives

This review has highlighted differences in the gut microbiota between captive and wild green sea turtles in the limited existing knowledge. Overall, Bacillota, Pseudomonadota, Bacteroidota and Verrucomicrobiota were identified among the most predominant bacterial phyla across all identified studies. Bacillota was identified as the most abundant phylum in captive turtles, but it ranked second to Pseudomonadota in wild turtles. Bacteroidota was prominent in the captive and wild groups, with relatively similar incidence across populations. Interestingly, Pseudomonadota was the dominant phylum in wild turtles, but it ranked lower in captive turtles. These findings emphasise the fact that the environmental conditions in which green sea turtles are located are critical factors in shaping the gut microbiome, and this has important implications for turtle health and conservation. Herein, we take a very preliminary step towards establishing consensus around what constitutes a normal, optimal gut microbiome for green sea turtles under captive or wild conditions. In order to better understand the gut microbial dynamics of green sea turtles throughout their life stages, future research on characterisation of microbial communities should focus on longitudinal studies that include detailed assessment of diet, and detailed descriptions of environmental conditions and/or habitat changes. Integrating multiomics approaches, such as metagenomics, metatranscriptomics, and metabolomics, alongside functional assays and advanced sequencing techniques, will deepen the understanding of microbial community functions and their role in host resilience. Well-designed comparative studies of captive and wild populations will be essential to identify context-specific microbiome patterns and provide critical insights for conservation strategies to mitigate the effects of environmental changes on this endangered species.

## Figures and Tables

**Figure 1 animals-15-01594-f001:**
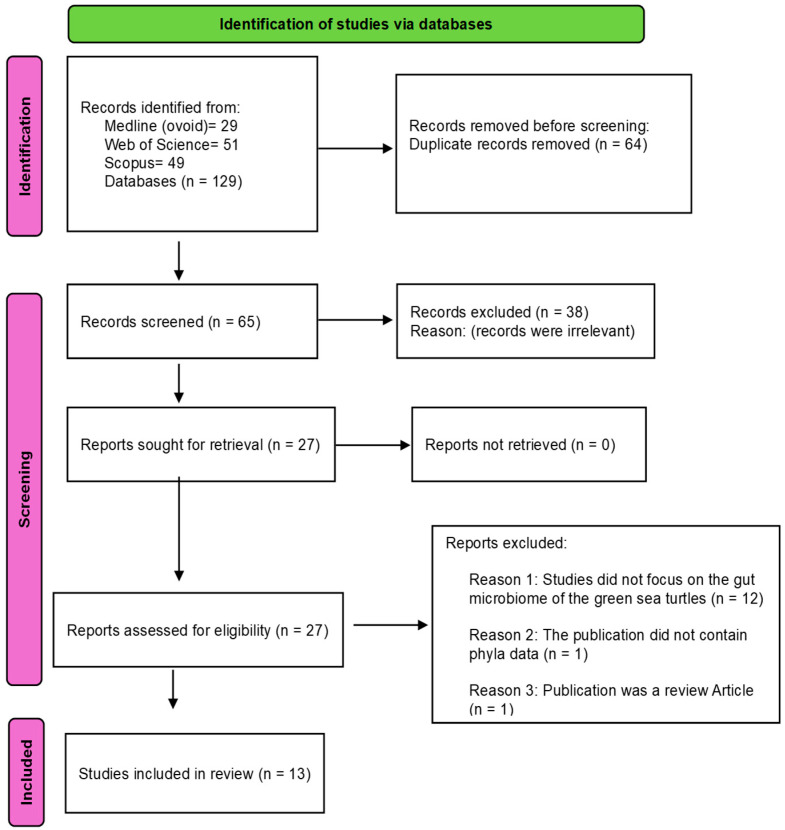
Methodological workflow of the study: Thirteen articles were selected for final analysis, all of which comprised original studies directly examining the gut microbiome of green sea turtles.

**Figure 2 animals-15-01594-f002:**
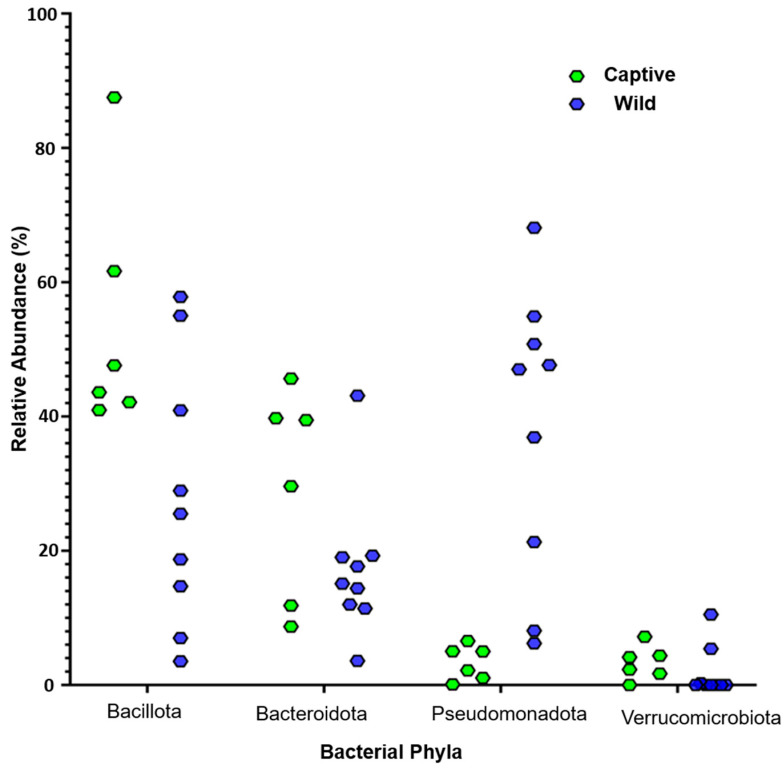
A scatter plot displaying the variability in the relative abundances of the top four dominant microbial phyla across the included studies. Green represents the relative abundance in captive green sea turtles, while blue represents the relative abundance in wild green sea turtles.

**Figure 3 animals-15-01594-f003:**
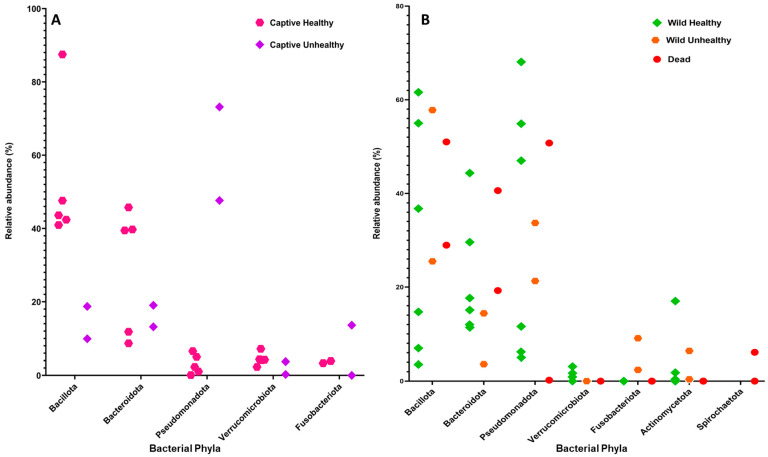
Relative abundance of bacterial phyla in the gut microbiome of green sea turtles, categorised by health status and environment. (**A**) Relative abundances of dominant and minor bacterial phyla in captive green sea turtles, with pink indicating healthy individuals and purple representing unhealthy individuals. (**B**) Relative abundances of distinct bacterial phyla in wild green sea turtles, with green representing healthy individuals, orange indicating unhealthy individuals, and red corresponding to dead individuals.

**Table 1 animals-15-01594-t001:** Summary of sea turtle gut microbiome of green sea turtle studies by location.

Region	Ranking of Top Four Dominant Phyla		References
Australia	Captive (Two studies, *n* = 20)	Wild (Four studies, *n* = 30)	[11,17,21,31,32]
	Bacillota Bacteroidota Verrucomicrobiota Pseudomonadota	Pseudomonadota Bacteroidota Bacillota Verrucomicrobiota	
USA	Captive (No reported study)	Wild (Three studies, *n* = 31)	[19,33,34]
	N/A	Pseudomonadota Bacteroidota Bacillota Verrucomicrobiota	
China	Captive (Three studies, *n* = 31)	Wild (No reported study)	[35,36,37]
	Bacillota Bacteroidota Verrucomicrobiota Pseudomonadota	N/A	
Brazil	Captive (One study, *n* = 8)	Wild (One study, *n* = 16)	[20]
	Bacteroidota Bacillota Verrucomicrobiota Pseudomonadota	Bacteroidota Bacillota Pseudomonadota Verrucomicrobiota	
Guinea-Bissau	Captive (No reported study)	Wild (One study, *n* = 7)	[12]
	N/A	Pseudomonadota Bacteroidota Bacillota Verrucomicrobiota	

Note: *n* represents the total number of turtles (samples) analysed for each subgroup and N/A indicates data not available.

## Data Availability

The original contributions presented in this study are included in the article.

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
