# Peer review of "Characterisation of the Gastrointestinal Microbiome of Green Sea Turtles (Chelonia mydas): A Systematic Review"

_animals, 2025, doi:10.3390/ani15111594_

Round 1
Reviewer 1 Report
Comments and Suggestions for Authors
Reviewer Edits for “Characterisation of the Gastrointestinal Microbiome of Green Sea Turtles (Chelonia mydas): A Systematic Review”
The present article is a meta-analysis of previously published works looking at the gut microbiomes of green sea turtles. The authors performed a thorough search of the literature and had appropriate inclusion criteria for which articles they included in their review. They grouped data into different categories including geographical locations, captive/wild populations, age, date of study, and health status/medical interventions. No statistical analysis was performed due to the heterogeneity of the data and inability to make any conclusions about different factors. They name the top 4 phyla present across all studies and discuss why there may be differences between mostly the wild and captive populations, which were mostly valid conclusions that have support from many studies. However, my biggest concern is that some of the data for wild populations were taken from sick or dead animals and some of the data for captive populations were from those receiving antimicrobial treatments. That data can still be looked at and discussed, but they most be delineated in an easy to understand format as those data points do not represent the authors main goals of trying to establish what “normal” flora looks like between populations of turtles. They should not be grouped together with all other healthy turtle data and then be used to make conclusions that wild turtles have a higher proportion of Pseudomondota phyla, as that may not be accurate if those samples are removed. My recommendation is to revise the manuscript with additional ranges for each phyla according the different health status or delineate on the scatter plot which samples are from those that were not considered healthy or have a second graphic that can help explain any differences.
Line 18: I would advise to use another word besides “putative” in the simple summary. It is not a very commonly used word.
Line 133-138: Just a copyediting issue, but the font is different in this section
Table 1: All phyla in the table should be written in the same naming scheme. Firmicutes, Bacteroidetes, Verrucomicrobia, and Proteobacteria in first data cell for Australia should be changed to the current naming scheme of Bacillota, Bacteroidota, Verrucomicrobiota, and Pseudomonadota so as to not confuse readers and increase uniformity across studies.
Line 310-312: You cite Keenan, S.W.; Engel, A.S.; Elsey, R.M. The alligator gut microbiome and implications for archosaur symbioses. Sci. Rep. 2013, 3, 2877. In reference to Akkermansia sp. (or Verrucomicrobiota). I could find no mention of either Verrucomicrobiota or Akkermasia throughout the paper when I searched it. Is this source correct? The previous source Rawski et al., briefly mentioned the part of enrichment with Akkermansia could induce gut inflammation in mice. However, if you are going to say “several studies” have suggested this, then you need to back it up with sources. Most studies actually show that Akkermasia is usually a beneficial gut bacteria in mammals and reduces gut inflammation, so make sure you are actually reporting both sides of the data and not cherry-picking. I will agree that you can say “nothing is known about its pathogenicity in reptiles.”
See following sources:
Xu Y, Duan J, Wang D, Liu J, Chen X, Qin XY, Yu W. Akkermansia muciniphila Alleviates Persistent Inflammation, Immunosuppression, and Catabolism Syndrome in Mice. Metabolites. 2023 Jan 28;13(2):194. doi: 10.3390/metabo13020194. PMID: 36837813; PMCID: PMC9961567.
Rodrigues VF, Elias-Oliveira J, Pereira ÍS, Pereira JA, Barbosa SC, Machado MSG, Carlos D. Akkermansia muciniphila and Gut Immune System: A Good Friendship That Attenuates Inflammatory Bowel Disease, Obesity, and Diabetes. Front Immunol. 2022 Jul 7;13:934695. doi: 10.3389/fimmu.2022.934695. PMID: 35874661; PMCID: PMC9300896.
Major issue: You really should not be including samples from sick, dead, or those receiving antimicrobial therapies in your sample set and not delineating them out somehow for your descriptive statistics and graphs. Those samples do not represent a normal population and if you are going to include them in your phyla groupings they need to be highlighted in some way or just not used. They can still be talked about in the discussion as having higher numbers of whatever taxa groups, but to include them and then try to compare healthy captive to a mixed population of healthy/unhealthy wild turtles is fundamentally flawed.
Author Response
Dear Editor and Reviewers,
We sincerely thank you for your thoughtful and constructive feedback on our manuscript. We appreciate the time and effort you dedicated to reviewing our work. We have carefully considered each comment and have revised the manuscript accordingly. Below, we provide a point-by-point response to each comment, along with a description of the changes incorporated in the revised manuscript.
Manuscript ID: animals-3552361
Manuscript Type: Systematic Review
Title: Characterisation of the Gastrointestinal Microbiome of Green Sea Turtles (Chelonia mydas): A Systematic Review
Authors: Dawood Ghafoor, Orachun Hayakijkosol, Carla Ewels, Robert Kinobe
Journal: Animals (MDPI)
Date: 14-4-2025
Reviewer 1:
Comments and Suggestions for Authors
Reviewer Edits for “Characterisation of the Gastrointestinal Microbiome of Green Sea Turtles (Chelonia mydas): A Systematic Review”
Comment 1: The present article is a meta-analysis of previously published works looking at the gut microbiomes of green sea turtles. The authors performed a thorough search of the literature and had appropriate inclusion criteria for which articles they included in their review. They grouped data into different categories including geographical locations, captive/wild populations, age, date of study, and health status/medical interventions. No statistical analysis was performed due to the heterogeneity of the data and inability to make any conclusions about different factors. They name the top 4 phyla present across all studies and discuss why there may be differences between mostly the wild and captive populations, which were mostly valid conclusions that have support from many studies. However, my biggest concern is that some of the data for wild populations were taken from sick or dead animals and some of the data for captive populations were from those receiving antimicrobial treatments. That data can still be looked at and discussed, but they most be delineated in an easy to understand format as those data points do not represent the authors main goals of trying to establish what “normal” flora looks like between populations of turtles. They should not be grouped together with all other healthy turtle data and then be used to make conclusions that wild turtles have a higher proportion of Pseudomondota phyla, as that may not be accurate if those samples are removed. My recommendation is to revise the manuscript with additional ranges for each phyla according the different health status or delineate on the scatter plot which samples are from those that were not considered healthy or have a second graphic that can help explain any differences.
Response: Thank you for your insightful feedback. In response, we have now stratified the phylum-level data by health status for both captive and wild green sea turtles. To enhance interpretability, we have incorporated new scatter plots that clearly distinguish healthy, unhealthy, and dead green sea turtles (Figure 3). These updates are detailed in a newly added paragraph in Section 3.4 (lines 375-394) of the main text in manuscript. Given that most original studies did not report antimicrobial treatments or provide clear definitions regarding disease severity or health status, we have explicitly addressed this limitation in the revised manuscript under the limitations section (lines 410–415).
Comment 2: Line 18: I would advise to use another word besides “putative” in the simple summary. It is not a very commonly used word.
Response: Thank you for the suggestion. The word “putative” has been replaced with “potential” to improve clarity and accessibility for a broader audience.
Comment 3: Line 133-138: Just a copyediting issue, but the font is different in this section
Response: Thank you for pointing this out. The formatting issue has been corrected, and the font in this section has been standardised to match the rest of the manuscript.
Comment 4: Table 1: All phyla in the table should be written in the same naming scheme. Firmicutes, Bacteroidetes, Verrucomicrobia, and Proteobacteria in first data cell for Australia should be changed to the current naming scheme of Bacillota, Bacteroidota, Verrucomicrobiota, and Pseudomonadota so as to not confuse readers and increase uniformity across studies.
Response: Thank you for your helpful suggestion. The phylum names in Table 1 have been updated to the current nomenclature Bacillota, Bacteroidota, Verrucomicrobiota, and Pseudomonadota to ensure consistency and avoid confusion for readers.
Comment 5: Line 310-312: You cite Keenan, S.W.; Engel, A.S.; Elsey, R.M. The alligator gut microbiome and implications for archosaur symbioses. Sci. Rep. 2013, 3, 2877. In reference to Akkermansia sp. (or Verrucomicrobiota). I could find no mention of either Verrucomicrobiota or Akkermasia throughout the paper when I searched it. Is this source correct? The previous source Rawski et al., briefly mentioned the part of enrichment with Akkermansia could induce gut inflammation in mice. However, if you are going to say “several studies” have suggested this, then you need to back it up with sources. Most studies actually show that Akkermasia is usually a beneficial gut bacteria in mammals and reduces gut inflammation, so make sure you are actually reporting both sides of the data and not cherry-picking. I will agree that you can say “nothing is known about its pathogenicity in reptiles.”
See following sources:
Xu Y, Duan J, Wang D, Liu J, Chen X, Qin XY, Yu W. Akkermansia muciniphila Alleviates Persistent Inflammation, Immunosuppression, and Catabolism Syndrome in Mice. Metabolites. 2023 Jan 28;13(2):194. doi: 10.3390/metabo13020194. PMID: 36837813; PMCID: PMC9961567.
Rodrigues VF, Elias-Oliveira J, Pereira ÍS, Pereira JA, Barbosa SC, Machado MSG, Carlos D. Akkermansia muciniphila and Gut Immune System: A Good Friendship That Attenuates Inflammatory Bowel Disease, Obesity, and Diabetes. Front Immunol. 2022 Jul 7;13:934695. doi: 10.3389/fimmu.2022.934695. PMID: 35874661; PMCID: PMC9300896.
Response: Thank you for your valuable suggestions. The Keenan et al. (2013) reference has been removed, as it does not mention Akkermansia or Verrucomicrobiota. The text has been revised to reflect that Akkermansia is generally beneficial in mammals, though context-dependent effects exist (lines 313-315). We also note that its pathogenicity role in reptiles is still unknown. Additional references have been added to provide a balanced view.
Comment 6: Major issue: You really should not be including samples from sick, dead, or those receiving antimicrobial therapies in your sample set and not delineating them out somehow for your descriptive statistics and graphs. Those samples do not represent a normal population and if you are going to include them in your phyla groupings they need to be highlighted in some way or just not used. They can still be talked about in the discussion as having higher numbers of whatever taxa groups, but to include them and then try to compare healthy captive to a mixed population of healthy/unhealthy wild turtles is fundamentally flawed.
Response: Your comments were highly appreciated and have been carefully considered in our revisions. We fully acknowledge the importance of distinguishing between healthy, sick, and deceased individuals. As information on antimicrobial treatment was not available in most of the included studies, and thus could not be used to separate those samples, we have addressed this issue explicitly in the limitations section. In response to your comment, we have revised our analysis to clearly separate the relative abundances of healthy, sick, and dead turtles. A new paragraph has been added to the results section (section 3.4) to discuss these distinctions, and new scatter plots have been incorporated to visually illustrate differences across health statuses.
Reviewer 2 Report
Comments and Suggestions for Authors
I recommend rejecting the manuscript at this stage.
Comment:
Lines 23-30: Comparison of the top four bacterial phyla revealed that Bacillota was the most abundant phyla in captive turtles (40.9%–87.5%), but it only ranked second (3.5%–57.8%) in wild turtles. Bacteroidota had comparable relative abundance in captive (8.7%–45.6%) and wild (3.6%–43.1%) populations. By contrast, the relative abundance of Pseudomonadota was higher in wild turtles (6.2% – 68.1%) compared to the captive population (0.1%–6.6%). Verrucomicrobiota was less prevalent in wild and captive populations with relative abundances ranging from 0.28–5.4%, and 2.3%–7.2% respectively.
The authors need to conduct an additional inventory of the published data. The authors indicate: "All used databases were searched from the time of inception to 25 April 2024". But the most important publication was published on 30 May 2024. In particular, in 2024, the article "Comparison of the intestinal flora of wild and artificial breeding green turtles (Chelonia mydas)" (Niu et al., 2024) was published, in particular, the authors indicate in the abstract: "Therefore, this study compared the gut microbiological characteristics of wild and artificially bred green turtles (Chelonia mydas) through high-throughput Illumina sequencing technology. The α-diversity of intestinal bacteria in wild green turtles, as determined by Shannon and Chao indices, significantly surpasses that of artificial breeding green turtles (p < 0.01). However, no significant differences were detected in the fungal α-diversity between wild and artificially bred green turtles. Meanwhile, the β-diversity analysis revealed significant differences between wild and artificially bred green turtles in bacterial and fungal compositions. The community of gut bacteria in artificially bred green turtles had a significantly higher abundance of Fusobacteriota including those belonging to the Paracoccus, Cetobacterium, and Fusobacterium genera than that of the wild green turtle. In contrast, the abundance of bacteria belonging to the phylum Actinobacteriota and genus Nautella decreased significantly. Regarding the fungal community, artificially bred green turtles had a significantly higher abundance of Fusarium, Sterigmatomyces, and Acremonium and a lower abundance of Candida and Rhodotorula than the wild green turtle. The PICRUSt2 analyzes demonstrated significant differences in the functions of the gut bacterial flora between groups, particularly in carbohydrate and energy metabolism. Fungal functional guild analysis further revealed that the functions of the intestinal fungal flora of wild and artificially bred green turtles differ significantly in terms of animal pathogens-endophytes-lichen parasites-plant pathogens-soil saprotrophswood saprotrophs. BugBase analysis revealed significant potential pathogenicity and stress tolerance variations between wild and artificially bred green turtles. Collectively, this study elucidates the distinctive characteristics of gut microbiota in wild and artificially bred green turtles while evaluating their health status. These findings offer valuable scientific insights for releasing artificially bred green turtles and other artificially bred wildlife into natural habitats.
Reference:
Niu X, Lin L, Zhang T, An X, Li Y, Yu Y, Hong M, Shi H and Ding L (2024) Comparison of the intestinal flora of wild and artificial breeding green turtles (Chelonia mydas). Front. Microbiol. 15:1412015. doi: 10.3389/fmicb.2024.1412015
Author Response
Reviewer 2:
Comments and Suggestions for Authors
I recommend rejecting the manuscript at this stage.
Comment:1 Lines 23-30: Comparison of the top four bacterial phyla revealed that Bacillota was the most abundant phyla in captive turtles (40.9%–87.5%), but it only ranked second (3.5%–57.8%) in wild turtles. Bacteroidota had comparable relative abundance in captive (8.7%–45.6%) and wild (3.6%–43.1%) populations. By contrast, the relative abundance of Pseudomonadota was higher in wild turtles (6.2% – 68.1%) compared to the captive population (0.1%–6.6%). Verrucomicrobiota was less prevalent in wild and captive populations with relative abundances ranging from 0.28–5.4%, and 2.3%–7.2% respectively.
The authors need to conduct an additional inventory of the published data. The authors indicate: "All used databases were searched from the time of inception to 25 April 2024". But the most important publication was published on 30 May 2024. In particular, in 2024, the article "Comparison of the intestinal flora of wild and artificial breeding green turtles (Chelonia mydas)" (Niu et al., 2024) was published, in particular, the authors indicate in the abstract: "Therefore, this study compared the gut microbiological characteristics of wild and artificially bred green turtles (Chelonia mydas) through high-throughput Illumina sequencing technology. The α-diversity of intestinal bacteria in wild green turtles, as determined by Shannon and Chao indices, significantly surpasses that of artificial breeding green turtles (p < 0.01). However, no significant differences were detected in the fungal α-diversity between wild and artificially bred green turtles. Meanwhile, the β-diversity analysis revealed significant differences between wild and artificially bred green turtles in bacterial and fungal compositions. The community of gut bacteria in artificially bred green turtles had a significantly higher abundance of Fusobacteriota including those belonging to the Paracoccus, Cetobacterium, and Fusobacterium genera than that of the wild green turtle. In contrast, the abundance of bacteria belonging to the phylum Actinobacteriota and genus Nautella decreased significantly. Regarding the fungal community, artificially bred green turtles had a significantly higher abundance of Fusarium, Sterigmatomyces, and Acremonium and a lower abundance of Candida and Rhodotorula than the wild green turtle. The PICRUSt2 analyzes demonstrated significant differences in the functions of the gut bacterial flora between groups, particularly in carbohydrate and energy metabolism. Fungal functional guild analysis further revealed that the functions of the intestinal fungal flora of wild and artificially bred green turtles differ significantly in terms of animal pathogens-endophytes-lichen parasites-plant pathogens-soil saprotrophswood saprotrophs. BugBase analysis revealed significant potential pathogenicity and stress tolerance variations between wild and artificially bred green turtles. Collectively, this study elucidates the distinctive characteristics of gut microbiota in wild and artificially bred green turtles while evaluating their health status. These findings offer valuable scientific insights for releasing artificially bred green turtles and other artificially bred wildlife into natural habitats.
Reference:
Niu X, Lin L, Zhang T, An X, Li Y, Yu Y, Hong M, Shi H and Ding L (2024) Comparison of the intestinal flora of wild and artificial breeding green turtles (Chelonia mydas). Front. Microbiol. 15:1412015. doi: 10.3389/fmicb.2024.1412015
Response: Thank you for your comments, which have helped improve the manuscript. We acknowledge the importance of the recent study by Niu et al. (2024), which was published after our search cut-off date of 25 April 2024. As this manuscript is a systematic review, our search strategy was time-bound to ensure a transparent and replicable methodology, in accordance with PRISMA guidelines. The selective addition of work by Niu et al at this stage would go against established guidelines for publishing systematic reviews. This would require a whole new methodology, re-analysis of all data and rewriting the entire manuscript. We have now acknowledged this limitation explicitly in the revised manuscript (Limitation section, lines 415-417), stating that newly published studies beyond our cut-off date may provide additional insights and should be considered in future reviews.
Reviewer 3 Report
Comments and Suggestions for Authors
This is an interesting and generally well-written manuscript. The authors have been transparent about the way they chose the studies that they decided to include. They also have been forthcoming about the limitations of the study and suggested directions for future research. They have demonstrated that there is a difference between the gut microbiome in captive and wild green sea turtles although in a sort of preliminary way, focusing on the different phyla. There are a couple suggestions that might be beneficial to readers. When they introduce the fact that this review is based on 13 separate studies, they may also want to mention the total number of samples that this includes. They also classify the turtles as juvenile, sub-adult or adult, perhaps including some further information (if possible) about what these classifications mean about the size and potential age of the turtles. As much information as possible about the turtles should be included. Finally, in the captive turtle category, because they are being fed, was there any similarity in the diets that they received?
Author Response
Reviewer 3:
Comments and Suggestions for Authors
Comments 1: This is an interesting and generally well-written manuscript. The authors have been transparent about the way they chose the studies that they decided to include. They also have been forthcoming about the limitations of the study and suggested directions for future research. They have demonstrated that there is a difference between the gut microbiome in captive and wild green sea turtles although in a sort of preliminary way, focusing on the different phyla. There are a couple suggestions that might be beneficial to readers. When they introduce the fact that this review is based on 13 separate studies, they may also want to mention the total number of samples that this includes. They also classify the turtles as juvenile, sub-adult or adult, perhaps including some further information (if possible) about what these classifications mean about the size and potential age of the turtles. As much information as possible about the turtles should be included. Finally, in the captive turtle category, because they are being fed, was there any similarity in the diets that they received?
Response: Thank you for your thoughtful comments and helpful suggestions. we have now included the total number of turtle samples analysed across the 13 studies in Table 1, with separate number for captive and wild green sea turtles.
Regarding age classifications, we acknowledge the variability in how studies reported age or developmental stage. While some studies clearly classified individuals as juvenile, sub-adult, or adult, others provided ranges of curved carapace length (CCL) measurements without specifying age categories. We have added information in Supplementary File (Table S2), which summarises the classification used in each study, including reported terms as juvenile, sub-adult, or adult and corresponding CCL ranges where available.
Lastly, we have reviewed the available information on the diets of captive turtles. As most of the included studies focused on wild green sea turtles and only a few reported detailed dietary information for captive turtles, we have summarised the relevant dietary details in the Supplementary File (Table S2), where available.
Round 2
Reviewer 1 Report
Comments and Suggestions for Authors
The edits have addressed my previous concerns.
Reviewer 2 Report
Comments and Suggestions for Authors
No comment .